# Use of Force Feedback Device in a Hybrid Brain-Computer Interface Based on SSVEP, EOG and Eye Tracking for Sorting Items

**DOI:** 10.3390/s21217244

**Published:** 2021-10-30

**Authors:** Arkadiusz Kubacki

**Affiliations:** Institute of Mechanical Technology, Poznan University of Technology, ul. Piotrowo 3, 60-965 Poznań, Poland; arkadiusz.kubacki@put.poznan.pl

**Keywords:** electroencephalography, EEG, electrooculography, EOG, steady-state visual evoked potential, SSVEP, eye tracking, force feedback

## Abstract

Research focused on signals derived from the human organism is becoming increasingly popular. In this field, a special role is played by brain-computer interfaces based on brainwaves. They are becoming increasingly popular due to the downsizing of EEG signal recording devices and ever-lower set prices. Unfortunately, such systems are substantially limited in terms of the number of generated commands. This especially applies to sets that are not medical devices. This article proposes a hybrid brain-computer system based on the Steady-State Visual Evoked Potential (SSVEP), EOG, eye tracking, and force feedback system. Such an expanded system eliminates many of the particular system shortcomings and provides much better results. The first part of the paper presents information on the methods applied in the hybrid brain-computer system. The presented system was tested in terms of the ability of the operator to place the robot’s tip to a designated position. A virtual model of an industrial robot was proposed, which was used in the testing. The tests were repeated on a real-life industrial robot. Positioning accuracy of system was verified with the feedback system both enabled and disabled. The results of tests conducted both on the model and on the real object clearly demonstrate that force feedback improves the positioning accuracy of the robot’s tip when controlled by the operator. In addition, the results for the model and the real-life industrial model are very similar. In the next stage, research was carried out on the possibility of sorting items using the BCI system. The research was carried out on a model and a real robot. The results show that it is possible to sort using bio signals from the human body.

## 1. Introduction

Electroencephalography is currently becoming increasingly common. Scientific research features a trend of attempts to not only use it for studying the brain, but also for control. Brain-computer interfaces (BCIs) are used for this purpose and are becoming increasingly popular in scientific research. These interfaces can recognize certain commands directly from the brain [1] and perform intended, predefined actions. This allows for controlling objects, such as Personal Care Robots [2]. EEG-based interfaces are quickly developing thanks to the declining prices of headsets [3]. EEG is a noninvasive method intended for registering brain activity on the skull’s surface with the use of electrodes. The number of possible commands and the effectiveness of their detection for a single BCI method are limited. Hence, hybrid BCIs are becoming increasingly popular [4]. Their classification is presented in Figure 1.

The number of classification commands is increased thanks to the use of a larger number of interfaces. In reality, BCIs are becoming increasingly combined with other interfaces, such as voice recognition or electromyography (EMG) [5,6,7,8,9]. It is possible to combine the Steady-State Visual Evoked Potential (SSVEP) with the P300 event-related potential [4], electromyography (EMG) with electroencephalographic activity (EEG) [10,11,12,13], or SSVEP with the ERD/ERS method [14,15,16]. An increasing number of hybrid BCI system solutions currently go beyond combining solely EEG-based systems or other signals derived from the human organism. The hybrid systems are being combined, for example, with visual systems [17]. There are also systems based on the recognition of artifacts derived from facial expressions and the movement of the eyeballs [18]. Such systems are most often used for controlling external devices. In the study [19], it is proposed controlling the wheelchair by means of a brain-computer interface based on the P300. They achieved a control efficiency of over 99%. The authors in the article [20] created a virtual platform for training disabled people. They were supposed to move a virtual wheelchair in the environment using the interface-brain-computer. In the study [21] it is presented an environment based on the ROS system for controlling a wheelchair. Using the SSVEP method, they controlled the movements of the cart. In this case, the trolley was an autonomous element. Using 6 blinking fields, the user selected a previously saved destination on the map generated with the Simultaneous localization and mapping (SLAM) algorithm. The algorithm is responsible for creating and updating the map and tracking the agent. In the study [22] it is presented controlling a virtual robot around a virtual apartment with a system based on P-300. The stimulators were two tactile actuators. It is not only wheelchairs that are the target of BCI control. In the article [23] it is proposed a system based on SSVEP for controlling a mobile car. They presented car steering at a distance of 15 m. A similar brain-computer interface was presented by the authors of the article [24]. They also controlled a mobile car. This time the car was controlled via WiFi. The car was based on the Arduino kit. In the study [25], it is presented the control of a mobile car by means of a brain-computer interface based on artifact recognition. Other vehicles controlled by brain-computer interfaces are flying vehicles. The authors of the article [26] were interested in the problem of loss of attention while controlling an unmanned aerial vehicle. They built a system based on a brain-computer interface that prevents the operator from falling into a state of inattention. At this point, the system informs by restoring the state of attention. Christensen et al. [27] developed a drone control system based on Five Class MI and Filter Bank CSP. The article [28] shows various interfaces for controlling unmanned aerial vehicles in recent years. They show that the control of such vehicles is most often carried out using MI Tasks. Not only are vehicles controlled via brain-computer interfaces. Increasingly, the controlled element is a robotic arm [29]. In the article [30] the Achic et al. presented a system based on SSVEP for controlling an assistive robot arm. The robot arm was mounted on a wheelchair. In the study [31], it is presented a system based on Mi tasks and SSVEP for 3DOF robot control. In the article [32], the authors presented a system based on SSVEP, eye blinking for Real-Life Meal-Assist Robot Control. Steady-state visual evoked potentials (SSVEPs) from occipital channels were used to select the food per the user’s intention. Athanasiou et al. [33] presented the system for controlling an anthropomorphic robotic arm using MI tasks. The problem with such interfaces is the lack of assurance feedback. Much information can be displayed on a monitor, but this may disrupt the operation of SSVEP-based interfaces. When used in hybrid interfaces, sound signals can be burdensome for users in the long-term. In this paper, the author proposed a solution based on force feedback. The novelty presented in this article is the construction of a hybrid brain-computer system based on SSVEP, EOG, eye tracking, and force feedback. The additional feedback significantly improves the results.

## 2. Materials and Methods

### 2.1. System Overview

The author built a hybrid brain-computer interface for controlling a six-axis industrial robot. The system is based on an EEG set and a visual system. Additional novelty was introduced with the use of force feedback on the robot’s position. The feedback module is described in the latter part of the paper. The robot’s movement takes place in three axes. Electrooculography (EOG) is used to move the robot, whereas the signals are collected from the EEG set. EOG is a noninvasive method of acquiring information about the resting potential near the eyeballs. Due to its structure, an eye is an electric dipole, consequently, it is possible to test such potentials in its vicinity.

Only EOG-based systems exist. In the article [34], Wang et al. proposed a system for controlling a wheelchair based on this system and new model of electrode. A leftward eyeball motion caused the robot arm to move in a positive direction in the given axis. A rightward eyeball motion caused the robot arm to move in a negative direction. Switching the selected axis was implemented using one of two methods studied in the latter part of the paper, i.e., by blinking twice and via a method based on the Steady-State Visually Evoked Potential (SSVEP). The diagram of the entire hybrid interface is presented in Figure 2. The hybrid brain-computer system is an online system. The program responsible for connecting all elements was written in C++. The program is responsible both for collecting information from other systems and for operating a feedback device. It implements API support from Emotiv EPOC+. The same program sends and receives information about the position of the robot’s tip. It communicates with the openVibe software via the VRPN server. In this way, it sends information about the start and end of the test, and receives information about the detection of one of the lamps blinking. Using Shered Memory, it communicates with the software responsible for operating the camera with the implemented eyeLike library. As in the previous case, the program sends information about the start and end of the test, and receives information about the current location of the eyeball centers. Also, through Shared Memory, it communicates with the software responsible for operating the industrial robot. Information is exchanged about the current position of the robot’s tip and the given position. The written program also communicates via UART with the controller of the feedback device. The software block diagram is presented in Figure 3.

### 2.2. EEG System

The tests were conducted with the use of the EPOC+ EEG set from Emotiv. The cap features 14 built-in electrodes + 2 reference electrodes. The electrodes are soaked in a saline solution before testing. A 16-bit ADC transducer is responsible for changing the electric signal into digital information. The frequency response is in the range from 0.16 to 43 Hz. In addition, a filter is implemented at 50 and 60 Hz. The EEG artefacts deriving from eyeball movement were detected by using a built-in smart module implemented in the API provided by the manufacturer based on a simplified EOG test. The author wrote software that implemented axis selection via the SSVEP.

### 2.3. Steady-State Visual Evoked Potential (SSVEP)

Steady-State Visual Evoked Potential (SSVEP) is a method used in brain-computer interfaces. It relies on a periodic measurement of evoked potentials generated by a human in response to recurring visual stimulation. When using a brain-computer interface based on such stimulation, the EEG displays a signal with a frequency corresponding to the excitement and its harmonics [35]. It is assumed that the frequency should be higher than 10 Hz [36]. The commonly used stimulating elements include devices relying on flashing light, such as an LED diode, flashing markers, or monitors with flashing chessboard patterns [37]. This method can be used on a person tested without prior training. The preliminary tests featured verification of the correct performance of the SSVEP method. Figure 2 presents the Fast Fourier Transform (FFT) of a signal collected from an electrode named Oz. Figure 4a presents the FFT when the LED diode is not flashing. Figure 4b presents the FFT of an output signal for the same electrode when the white LED diode is flashing with the frequency of 15 Hz.

The program to handle SSVEP was written in the openVibe software. The algorithm uses a spatial filter to select electrodes. The signal then passes through the Bandpass Butterworth 4th order filter. The time-based epoching block breaks the signal into blocks. The signal then goes to the feature aggregator block. A previously trained SVM was used as the classifier. The same algorithm was used to learn the classifier. The signal flow diagram is shown in Figure 5

The SSVEP Signal Model can be represented as:(1)$$y_{i}\left(t\right)=\overset{N_{h}}{\underset{k=1}{\sum }}\left(a_{i,k}\sin2\pi kft+b_{i,k}\cos\sin2\pi kft\right)+E_{i,k}$$
where *f* is the frequency, *k* is the harmonics, $a_{i,k}\;$ and $b_{i,k}\;$ is the amplitude and $E_{i,k}$ is the noise and artifacts. 

### 2.4. Force Feedback Device

Since the robot’s tip is controlled via eyeballs, it is not always within the user’s field of view. The related errors are especially visible when manipulating the robot’s tip near the obstacle. The author decided to introduce force feedback. Power feedback is applicable in other areas [38]. The purpose of the device was to invoke the impression of force proportional to the signal from the robot’s tip. The signal can be derived from a force sensor, if the test requires the recognition of force at the robot’s tip or from a distance sensor, if the test requires the robot to be positioned without touching the surrounding elements. The microcontroller collects data from the strain gauge bridge or distance sensor mounted at the robot’s tip and it controls the servo in such a way that it moves the moving block. Pulling out the block causes pressure on the skin of the examined person. 

The developed system also enables using the current position collected from the robot’s controller. For this purpose, a prototype force feedback device mounted on the user’s shoulder was designed and built (Figure 6). In the case of this article, the pressure of the movable block on the user’s skin is inversely proportional to the distance between the robot tip and the obstacle (Figure 7). The closer the tip is to the obstacle, the more pressure the device exerts on the user’s skin.

### 2.5. Mitsubishi RV-12sl Industrial Robot in the Simulation Environment 

The mathematical and graphical model was implemented in a Unity 3D environment. Mitsubishi RV-12sl is a six-axis industrial robot. A table of parameters D-H, presented in Table 1, was developed for the implementation of inverse kinematics. The parameters specifying the robot’s overall dimensions are presented in Figure 8.

The inverse kinematics model was presented by the author in a previous article [39]. 

### 2.6. Changing the Robot’s Active Axis with Methods Utilising EEG and SSVEP Artefacts

The axle was changed using the SSVEP system. It utilized 3 lamps flashing at the frequency of 15 Hz for axis X, 17 Hz for axis Y and 19 Hz for axis Z. Changing the active axis is separate from the movement of the robot arm. In this version, the system cannot simultaneously control the robot arm and change the active axis. It is impossible to select the active axis and see the working area and vice versa at the same time. To change the axis after the move was made, the user had to turn his head to the panel with the flashing lights. An alternative to this solution was to blink the eyeballs to change the axis to the next one. Tests showed, however, that this method is failing to detect errors caused by involuntary blinking of the subjects.

### 2.7. Comparison of the Accuracy of Robot Model Tip Positioning with and without Feedback

Test featured a comparison of the accuracy of the Mitsubishi RV-12sl virtual robot tip positioning with the use of feedback. Three people aged 30–40 took part in the research. All of the test takers do not have problems with the brain or diseases. On the day of the examination, they were refreshed and felt no discomfort. The test site was separated from the noise. The test subject was instructed to move the tip towards the tip of a virtual box within the shortest time possible. A combined 50 attempts were conducted for the enabled and disabled feedback module. The test station consisted of the Emotiv EPOC+ cap specified in the previous paragraph, feedback module mounted on the test subject’s shoulder, and the kinematic model of the Mitsubishi RV-12sl industrial robot. A module mounted on the test subject’s shoulder was used to obtain feedback. The robot’s graphical model and inverse kinematics were implemented in the Unity environment. The software transmitted information about the robot’s tip distance from the target cube to the feedback module via a RS-232 interface. This allowed for the simulation of the distance sensor’s operation in 3 axles. The virtual test station’s view in the program is presented in Figure 9.

The next test featured a comparison of the accuracy of the Mitsubishi RV-12sl real-life robot tip positioning with the use of feedback (Figure 10) as well as its comparison with the model’s results. Also, 50 attempts for the enabled and disabled feedback module were conducted.

### 2.8. Sort Items Using a Hybrid Brain-Computer Interface with Force Feedback Enabled

In the next test, the test person was tasked with sorting the balls using the tip of the robot. The box was randomly selected by the system. The selected box was marked with a green LED. In each sample, the test person was asked to sort 20 balls. A total of 60 trials were conducted. The lowering of the ball was done by blinking the eyelids twice. Correct and incorrect attempts were counted separately. In the case of incorrect attempts, the ball missed the box, and the boxes were mixed up. The tests were carried out on a robot model and then confirmed on a real robot. The robot was controlled in the same way as in the previous point. Two scenarios were created. In the first one, all the boxes were on the same height. The boxes were placed 0.7 m from the base of the robot. The distance between the centers of the boxes was 0.37 m. The size of the boxes is 0.1 × 0.1 × 0.1 m. There were no obstacles between the boxes. In the second scenario, the boxes were at different heights. The first box is 0.45 m high, the second is 0.7 m, and the third is 0.5 m. Additionally, there were obstacles above boxes 1 and 3. The distance between the centers of the boxes did not change. The same people described in the previous point participated in this study. The subjects conducted 20 trials each. Figure 11 presents the view of the robot model seen by the test subjects.

In the tests on the real robot, the same distances were kept as in the tests on the model. To maintain the same conditions, the user saw the image from two cameras placed in the vicinity of the robot. The image from the cameras is presented in Figure 12.

## 3. Results

### 3.1. Results of Tests Concerning the Accuracy of the Robot Model Tip Positioning with and without Feedback

The measurements from 50 attempts were superimposed on a chart after being collected. Figure 13 presents the results of virtual robot tip positioning both with enabled and disabled feedback. The results are presented in three plains. The next step featured statistical calculations which are presented in Table 2.

The simulation tests were repeated on the industrial robot during the next test. The measurements from 50 attempts were also superimposed on a chart after being collected. Figure 14 presents the results of the robot tip positioning both with feedback enabled and disabled. The results are presented in three plains. The next step featured statistical calculations which are presented in Table 3. A sample trajectory of the robot’s tip movement is presented in Figure 15.

### 3.2. Results of Sorting Elements Using a Hybrid Brain-Computer Interface with Force Feedback Turned on

During the research, runs from 60 trials were recorded. 20 balls were sorted for each trial. The number of correctly and incorrectly sorted balls was also recorded. Erroneous attempts were divided into errors resulting from missed boxes and balls thrown into the wrong box. Figure 16 shows the position of the robot tip for one of the trials. The test turned out to be not very complicated and after setting the arm at the appropriate height, the movement took place only in the *Y* axis. The next step featured statistical calculations which are presented in Table 4.

The simulation tests were repeated on the industrial robot during the next test. As in the previous test, the waveforms and information about the correctness of sorting for 60 trials were recorded. Due to the complexity of the test, the movement took place in more than one axis. The course of one of the tests is shown in Figure 17. The next step featured statistical calculations which are presented in Table 5.

## 4. Discussion

### 4.1. Discussion on the Results of Tests on the Accuracy of the Robot Model Tip Positioning with and without Feedback

Previous robot arm control systems in the literature focused on making a movement or selecting a predefined robot program. These articles did not focus on positioning the robot with the system. In this study, subjects must carefully inspect and stop the robotic arm as close to the box as possible. Therefore, the main goal of this work is to prove that a hybrid BCI can provide satisfactory precision in robotic arm control to cope with difficult daily tasks.

The presented results clearly demonstrate that substantially better results were obtained with feedback enabled. Only a few attempts without using feedback can be compared with attempts with feedback enabled. The average distance of the robot’s tip from the target is 4.5 times lower with feedback enabled than with feedback disabled. Similar results were obtained for the standard deviation. It is 5.5-times lower in favor of the test with feedback enabled. The time for the attempts with feedback enabled is approximately 19% lower than for the attempts with feedback disabled. 

The presented results for the real-life robot clearly demonstrate that substantially better results were obtained with feedback enabled, as was in the simulation tests. Similar to previous tests, most attempts with feedback disabled substantially deviated from attempts with feedback enabled. The average distance of the robot’s tip from the target is 4 times lower with feedback enabled than with feedback disabled. Similar results were obtained for the standard deviation. It is 3.9-times lower, in favor of the test, with feedback enabled. The time taken for attempts with feedback enabled is approximately 24% lower than for attempts with feedback disabled. 

A similar result can be observed when comparing the simulation test results with the real-life robot test results. In both cases, the results with enabled feedback are better than those with feedback disabled for both times and distances.

### 4.2. Discussion on the Results of Sorting Elements Using a Hybrid Brain-Computer Interface with Force Feedback Turned on

Most BCI systems only allow the selection of predefined programs. This article presents robot control in simulated working conditions consisting in sorting elements. Such systems may help people with disabilities in the future.

The results of the research on sorting on the virtual model are reflected in reality. Correct sort attempts were maintained at 90% for both tests. Most of the errors were due to the small size of the box and the user’s insufficient field of view. The time of the trials depended on the level of complexity of the traffic that had to be performed. Longer trials were observed when there were obstacles on the board and the boxes were at different heights from the ground. The test duration in the second scenario was 71% longer for the model and 78% longer for the real robot. Standard deviation was greater by about 30% in scenario number 2. Such results were obtained both on the model and the real robot. In both scenarios, the errors resulting from choosing the wrong box amount to around 3%. Most of the errors resulting from throwing the ball into the wrong container resulted from the accidental triggering of the throwing down trigger. There was a significant increase in errors in the second scenario resulting from missing the box. The number of failed attempts is approx. 70% higher. This situation results from covering part of the field of view with obstacles. The tests showed a low level of effectiveness when the samples were fully valid. Only approx. 25% of the trials for scenario number 1 and approx. 20% of trials for scenario number 2 were flawless. A much higher result was obtained for trials with a maximum of one error. In this case, it was approx. 50% of the trials for the scenario number 1 and approx. 40% of the trials for the scenario number 2. The results in both cases were lower for the scenario number 2. This was due to insufficient field of view by the user.

## 5. Conclusions

The built hybrid brain-computer system enables the sorting of objects using signals from the human body with a correctness of 90%. For better results, the author intends to focus on increasing the user’s field of view. Such systems can allow a disabled person to return to work.

## Figures and Tables

**Figure 1 sensors-21-07244-f001:**
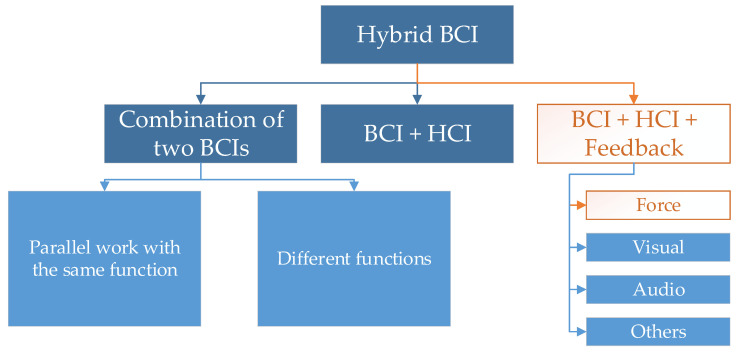
Classification of hybrid BCIs.

**Figure 2 sensors-21-07244-f002:**
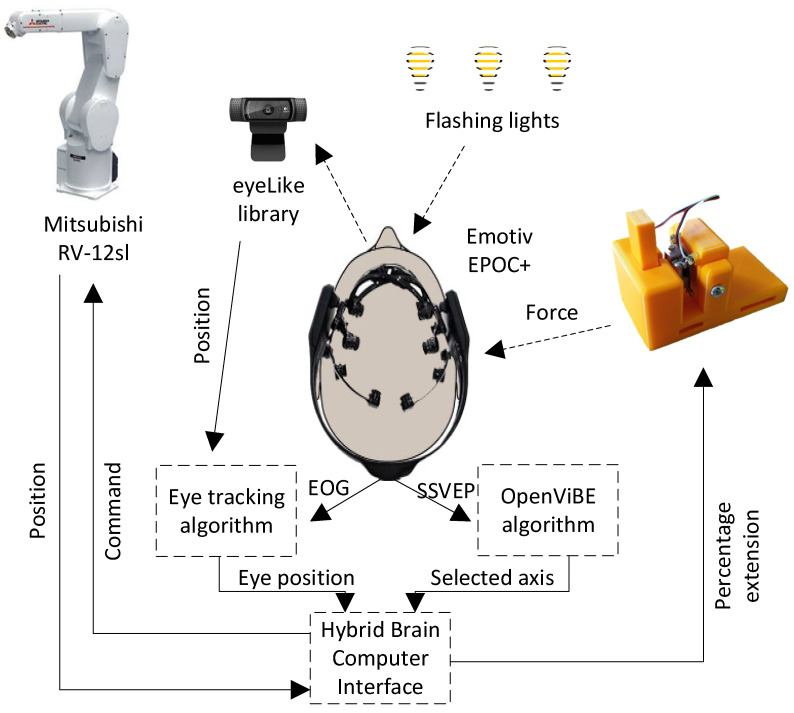
Hybrid brain-computer interface diagram.

**Figure 3 sensors-21-07244-f003:**
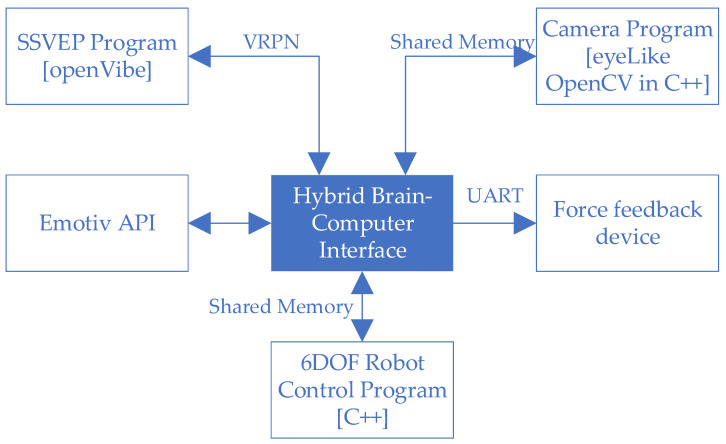
Structure of hybrid brain-computer interface software.

**Figure 4 sensors-21-07244-f004:**
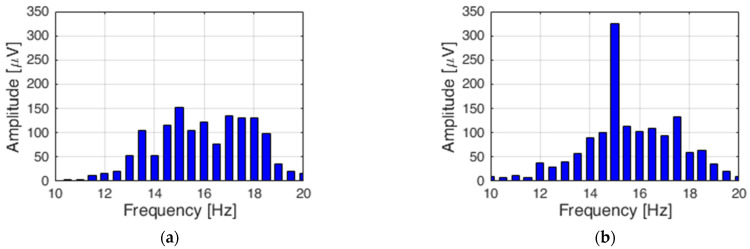
Electrode Oz output signal FFT for: no lamp (**a**) and a lamp flashing with 15 Hz (**b**).

**Figure 5 sensors-21-07244-f005:**
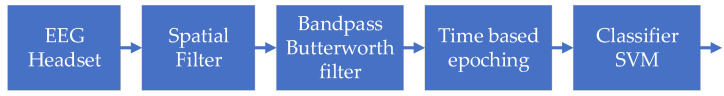
Data flow of SVEEP signal.

**Figure 6 sensors-21-07244-f006:**
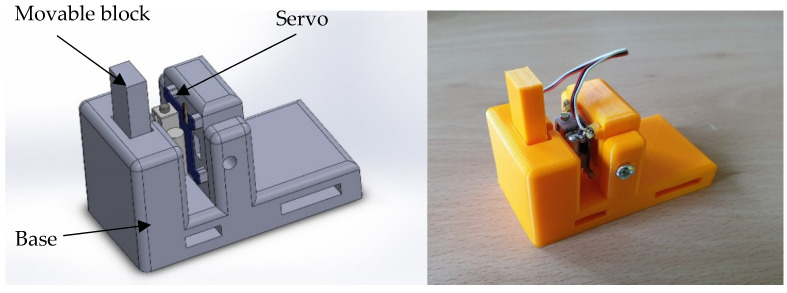
Force feedback device model and device.

**Figure 7 sensors-21-07244-f007:**
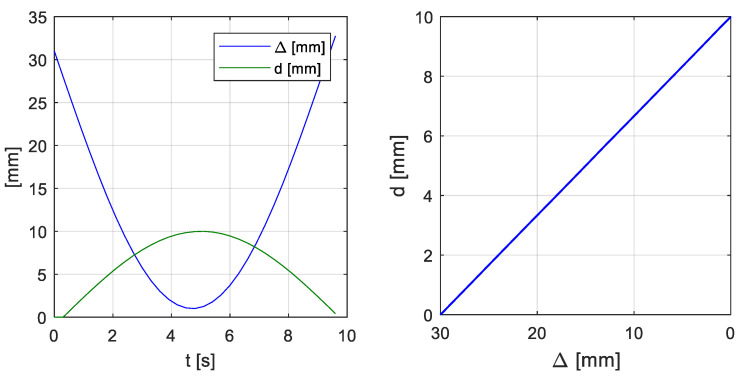
Plot of block extension and distance of robot tip from the box versus time (**left**) and plot of block extension versus distance of robot tip (**right**).

**Figure 8 sensors-21-07244-f008:**
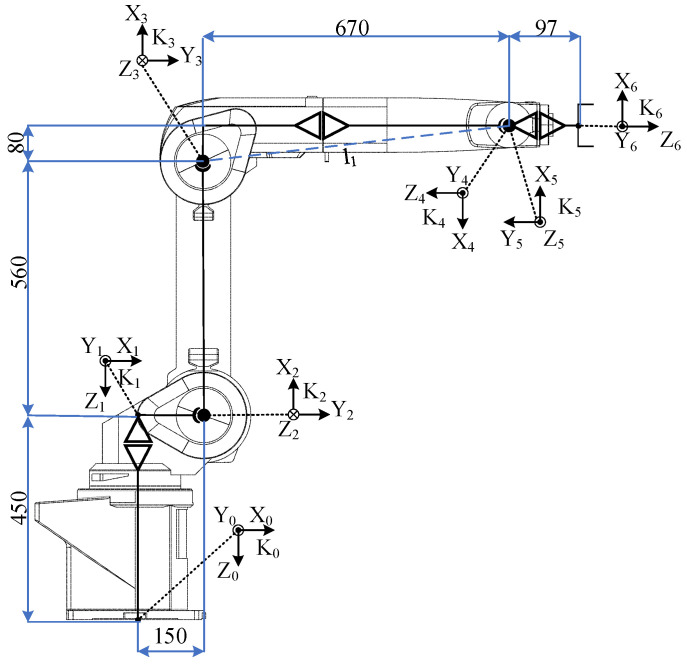
Mitsubishi RV-12sl industrial robot’s overall dimensions.

**Figure 9 sensors-21-07244-f009:**
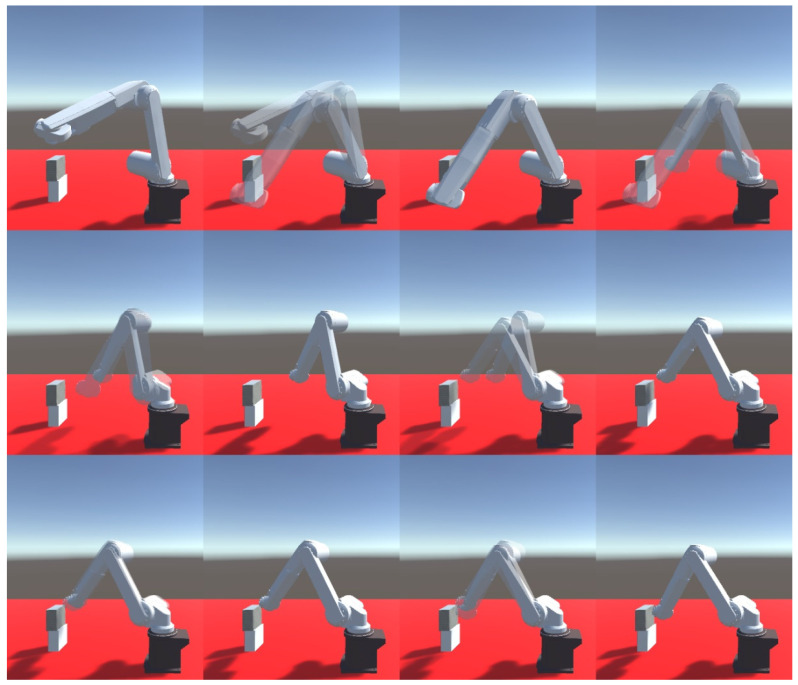
Sample robot movement in simulation environment during one of attempts.

**Figure 10 sensors-21-07244-f010:**
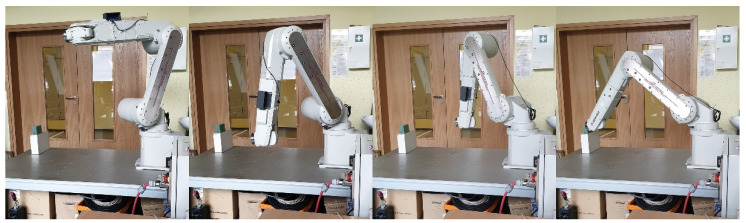
Sample robot movement in real-life station for force feedback testing during one of attempts.

**Figure 11 sensors-21-07244-f011:**
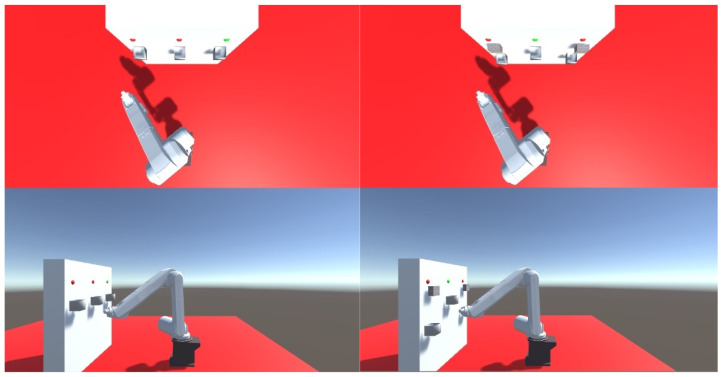
View of sort test performed on model. First scenario on left; second scenario on right.

**Figure 12 sensors-21-07244-f012:**
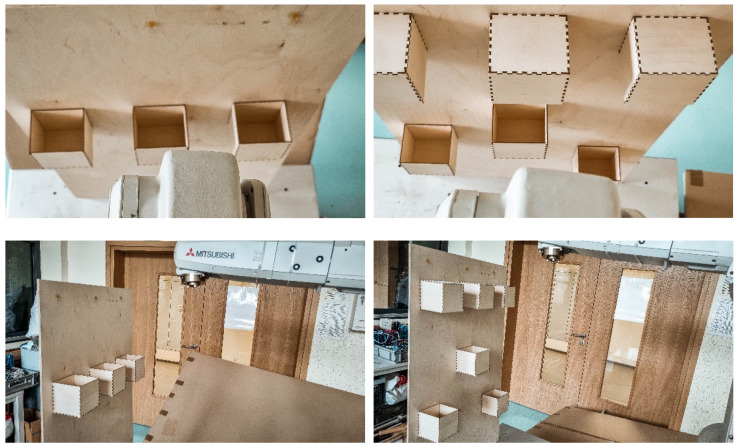
View of sorting test carried out on a real robot. First scenario on left; second scenario on right.

**Figure 13 sensors-21-07244-f013:**
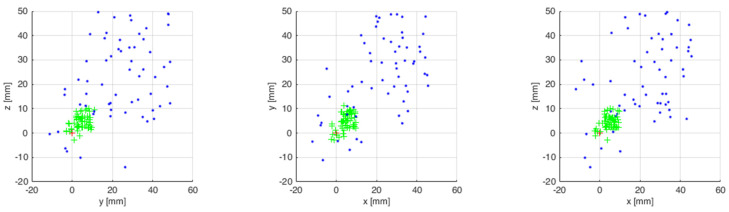
End points of virtual robot tip positioning with enabled and disabled feedback.

**Figure 14 sensors-21-07244-f014:**
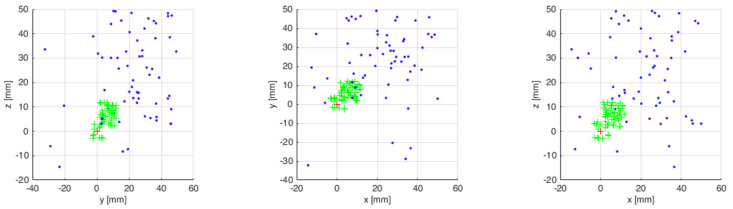
End points of robot tip positioning with enabled and disabled feedback.

**Figure 15 sensors-21-07244-f015:**
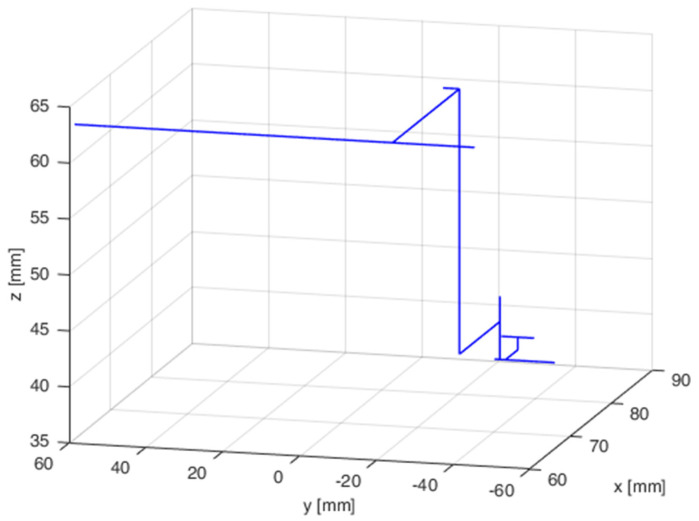
Sample trajectory of robot’s tip movement in one of attempts.

**Figure 16 sensors-21-07244-f016:**
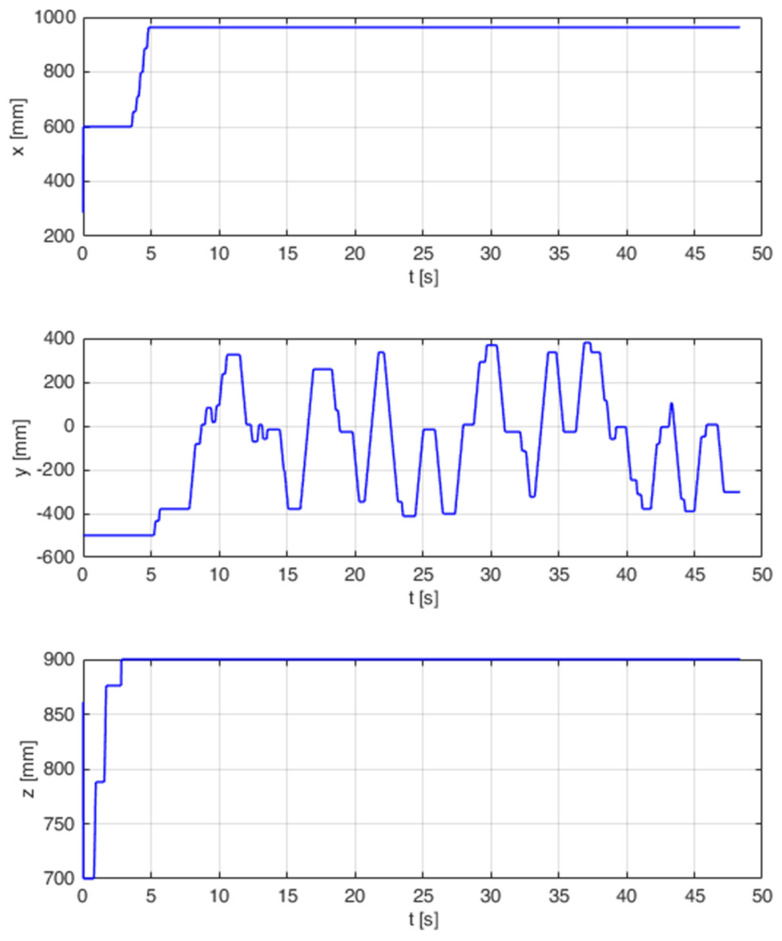
End points of virtual robot tip positioning endpoints when trying to sort.

**Figure 17 sensors-21-07244-f017:**
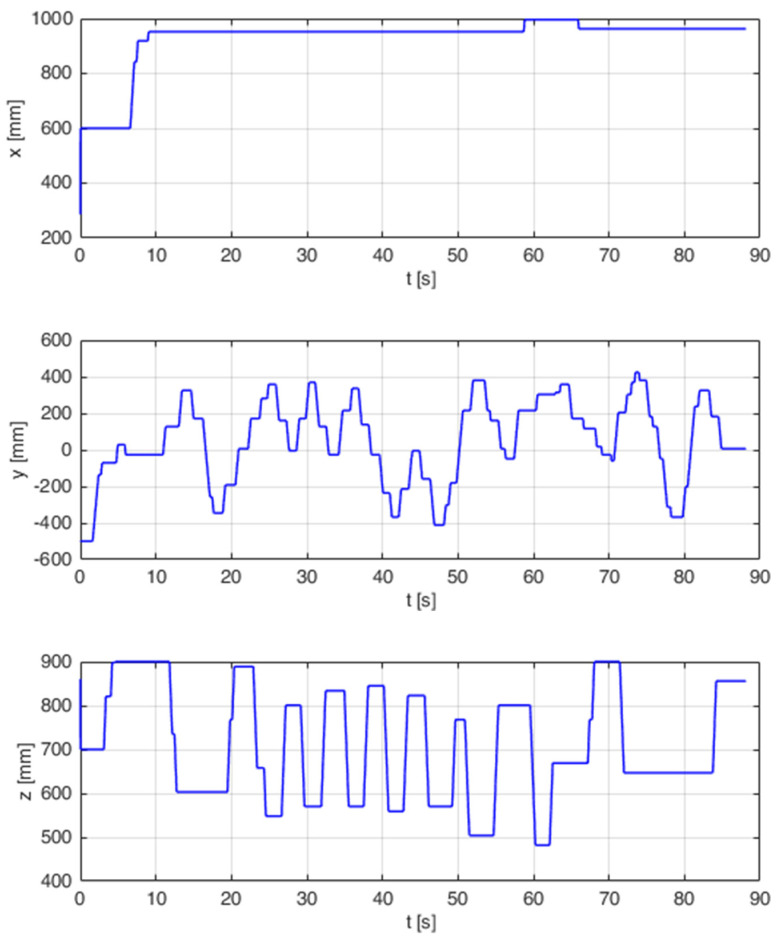
End points of real robot tip positioning endpoints when trying to sort.

**Table 1 sensors-21-07244-t001:** Table specifying Mitsubishi RV-12sl industrial robot’s parameters D-H.

i	a [mm]	α [rad]	d [mm]	θ [rad]
0	150	0	-	-
1	0	$\frac{\pi }{2}$	−450	$\theta_{1}$
2	560	0	0	$-\frac{\pi }{2}+\theta_{2}$
3	80	$\frac{\pi }{2}$	0	$\theta_{3}$
4	0	$-\frac{\pi }{2}$	−670	$\pi +\theta_{4}$
5	0	$\frac{\pi }{2}$	0	$\pi +\theta_{5}$
6	-	-	97	$\theta_{6}$

**Table 2 sensors-21-07244-t002:** Statistical results of a virtual robot’s tip positioning.

	No Feedback	Feedback
Average time	48.26 s	39.63 s
Standard deviation	10.12 s	9.83 s
Minimum time	30.60 s	20.40 s
Maximum time	64.40 s	54.30 s
Average distance	41.29 mm	9.00 mm
Standard distance deviation	13.12 mm	2.31 mm
Minimum distance	10.40 mm	4.48 mm
Maximum distance	65.58 mm	12.98 mm

**Table 3 sensors-21-07244-t003:** Statistical results of robot’s tip positioning.

	No Feedback	Feedback
Average time	49.35 s	38.36 s
Standard deviation	14.03 s	16.18 s
Minimum time	25.26 s	15.00 s
Maximum time	74.72 s	63.57 s
Average distance	39.71 mm	10.11 mm
Standard distance deviation	12.81 mm	3.2 mm
Minimum distance	9.99 mm	3.01 mm
Maximum distance	64.82 mm	15.31 mm

**Table 4 sensors-21-07244-t004:** Statistical results of virtual robot tip positioning endpoints when trying to sort.

	Model	Real Robot
Average time	58.18 s	56.87 s
Standard deviation	8.75 s	8.50 s
Minimum time	40.40 s	38.10 s
Maximum time	75.40 s	69.90 s
Incorrectly sorted	25 (2.1%)	31 (2.6%)
Missing boxes	56 (4.7%)	60 (5%)
Total wrong	81	91
Correctly sorted	1119 (93.2%)	1109 (92.4%)
Flawless trials	16 (26.7%)	13 (21.7%)
Attempts with a maximum of 1 failure	33 (55%)	30 (50%)

**Table 5 sensors-21-07244-t005:** Statistical results of robot tip positioning endpoints when trying to sort.

	Model	Real Robot
Average time	99.76 s	101.33 s
Standard deviation	11.51 s	11.27 s
Minimum time	80.70 s	70.90 s
Maximum time	118.80 s	129.20 s
Incorrectly sorted	38 (3.2%)	42 (3.5%)
Missing boxes	99 (8.25%)	102 (8.5%)
Total wrong	137	144
Correctly sorted	1063 (88.6%)	1056 (88%)
Flawless trials	12 (20%)	11 (18.3%)
Attempts with a maximum of 1 failure	25 (41.7%)	22 (36.7%)

## Data Availability

Not applicable.

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
