# Peer review of "Use of Force Feedback Device in a Hybrid Brain-Computer Interface Based on SSVEP, EOG and Eye Tracking for Sorting Items"

_sensors, 2021, doi:10.3390/s21217244_

Round 1
Reviewer 1 Report
The author proposes a system of a hybrid interface with BCI and eye tracking.
A unique point is the system has a force feedback to tell the interface's user a state of a robot arm.
The description of the protocol is insufficient.
The user of the proposed hybirid BCI/eye-tracking interface is required two cotrol---selecting an axis (out of three) and a direction (left or right) for the selected axis.
I cannot imagine the user control these two parameter simulutenously because the stimuli of the SSVEPs would be the out of the feild when the eye-ball moves to specify the direction.
Please add more description about it.
What kind of signal does the force feedback tell the user?
The author told that the force feedback is invoke the impression of force proportional.
But I cannot understand if this feedback helps the user to contraoll the arm exactly.
Please describe the detail the feedback signal.
What is SLAM (line 62, page 2)?
Figure 5 may not be reffered in the main text.
Author Response
Dear Reviewer,
I would like to thank you very much for your comments. I have modified the paper according to your remarks, as follows.
The description of the protocol is insufficient...
The description has been improved and expanded.
Changing the active axis is separate from the movement of the robot arm. In this version, the system cannot simultaneously control the robot arm and change the active axis. It is impossible to select the active axis and see the working area and vice versa at the same time. To change the axis after the move was made, the user had to turn his head to the panel with the flashing lights. An alternative to this solution was to blink the eyeballs in order to change the axis to the next one. Tests have shown, however, that this method is failing to detect errors caused by involuntary blinking of the subjects.
What kind of signal does the force feedback tell the user?
The description has been improved and expanded.
A prototype force feedback device is mounted on the user’s shoulder. In the case of this article, the pressure of the movable block on the user's skin is proportional to the distance of the robot tip from the obstacle. The closer the tip is to the obstacle, the more pressure the device exerts on the user's skin. A graph has been added to show the block extension depending on the distance of the robot tip from the obstacle (Fig.7).
What is SLAM (line 62, page 2)?
The description has been improved and expanded. Simultaneous localization and mapping (SLAM). The algorithm is responsible for creating and updating the map and tracking the agent. It was not sufficiently described in the previous version of the article.
Figure 5 may not be reffered in the main text.
The numbering of figures has been improved.
Author
Reviewer 2 Report
This manuscript proposed a hybrid brain-computer system based on the SSVEP. This method proposed by the authors is very reasonable, and detailed description and sufficient details are also given in the technical part. The composition of the article is clear and the writing is reasonable, which make readers can understand the technical details. The experimental results shows that the proposed method is obviously better than the traditional method. For these reasons, it is recommended to accept this manuscript.Author Response
Dear Reviewer,
I would like to thank you very much for your comments. I have modified the paper according to your remarks. The language in the article has also been improved.
Author
Reviewer 3 Report
In this manuscript the construction of a hybrid brain-computer system based on SSVEP, EOG, eye tracking and force feedback is proposed. The work is interesting and scientifically sound. Overall, it is a good-structured manuscript and in general well-written. There are a few parts throughout the manuscript that made it hard to follow the flow. Below are some comments to the author:
- It is deemed that the Abstract is of high importance in a paper since it gives a briefing of the work and prepares the reader for the manuscript. Thus, it should be excellent and strong. The repeated phrase “the author” ( i.e. the author proposed, the author tested, the author verified etc) renders the Abstract not well-written and does not leave the reader with a good feeling to continue to the manuscript. Please use passive voice instead and revise appropriately.
- The literature is outdated and only 2 studies is presented from 2021. Please include the following recent studies:
- Antoniou, E., Bozios, P., Christou, V., Tzimourta, K. D., Kalafatakis, K., G Tsipouras, M., ... & Tzallas, A. T. (2021). EEG-Based Eye Movement Recognition Using the Brain–Computer Interface and Random Forests. Sensors, 21(7), 2339.
- Ha, J., Park, S., Im, C. H., & Kim, L. (2021). A Hybrid Brain–Computer Interface for Real-Life Meal-Assist Robot Control. Sensors, 21(13), 4578.
- Wang, X., Xiao, Y., Deng, F., Chen, Y., & Zhang, H. (2021). Eye-Movement-Controlled Wheelchair Based on Flexible Hydrogel Biosensor and WT-SVM. Biosensors, 11(6), 198.
- Same with Comment 1 for lines 45-89 of the Introduction. The author could use passive voice and replace “the authors of the study ..” with “In the study .. it is proposed”. Please revise.
- The Discussion section is poor. The author should include the limitations of the current work and the advantages of the proposed system over other similar systems in the literature. For example Lines 87-89 “The novelty presented in this article is the construction of a hybrid brain-computer system based on SSVEP, EOG, eye tracking and force feedback. The additional feedback significantly improves the results” should be included and expanded in the Discussion. A comparison table is also highly recommended.
- The last paragraph of the manuscript (Lines 323-326) could be inserted in the separate section “Conclusions”.
Author Response
Dear Reviewer,
I would like to thank you very much for your comments. I have modified the paper according to your remarks, as follows.
- The text has been corrected as indicated.
- Literature expanded to include this year's entry. Numbers [18, 29, 32, 34]
- The text has been corrected as indicated.
- The section has been expanded. The results were compared in the text.
- The specified text range has been moved to a new paragraph.
Author
Round 2
Reviewer 1 Report
I recommend to change the title of the manuscript because the key idea of this research is with the force feedback system. I think it is better that the author insist this point in the title.
Author Response
Dear Reviewer,
I would like to thank you very much for your comments. I have modified the title according to your remarks.
Author
Reviewer 3 Report
The author performed the appropriate changes and improved the quality of the manuscript.
Author Response
Dear Reviewer,
I would like to thank you very much for your comments.
Author